# Improving the mental health and mental health support available to adolescents in out-of-home care via Adolescent-Focused Low-Intensity Life Story Work: a realist review

Simon P Hammond  ,[1,2] Ella Mickleburgh,[2] Claire Duddy  ,[3] Rachel Hiller,[4] Elsbeth Neil,[5] Rosie Blackett,[6] Kevin Williams,[7] Jon Wilson,[2] Geoff Wong[3]

For numbered affiliations see end of article.

**Correspondence to**
Dr Simon P Hammond;
s.hammond@uea.ac.uk

## ABSTRACT

**Objectives** Life Story Work (LSW) is used to promote the mental health and well-being of children and adolescents living in out-of-home care. LSW should be offered to all but is conventionally delivered in high-intensity ways. Low-intensity approaches are more accessible but there is significant variation and little guidance for supporting adolescents. We aimed to create guidance for Adolescent-Focused Low-Intensity LSW.

**Design** Realist review.

**Data sources** MEDLINE, Embase, PsycINFO, Sociology Collection (ProQuest), CINAHL, CDAS, Web of Science (SCIE, SSCI), Social Care Online and grey literature sources. Searches were performed between December 2021 and March 2022.

**Eligibility criteria** Documents on children and adolescents in care, LSW and/or low-intensity interventions to improve mental health were included. Documents focusing on parenting style and contact with birth family were excluded.

**Analysis** Documents were analysed using a realist logic of analysis. In consultation with Content Expert Groups (comprising professionals and care leavers), we developed an initial programme theory. Data relating to and challenging the initial programme theory were extracted and context-mechanism-outcome-configurations developed, critiqued and refined in an iterative fashion. Interpretations were drawn from context-mechanism-outcome-configurations to enhance the programme theory.

**Results** 75 documents contributed to the analysis. Generally, studies were small-scale and lacked in-depth methods and evaluation descriptions. Findings indicated important factors contribute to the development of high-quality Adolescent-Focused Low-Intensity LSW. Adolescent-Focused Low-Intensity LSW should be person-centred, begin in the now, involve co-construction, record everyday positive life events and be supported by trained carer(s). Context-mechanism-outcome-configurations relating to these themes are reported.

**Conclusions** Using this knowledge we developed initial practice guidance to support social care to deliver better quality Adolescent-Focused Low-Intensity LSW more consistently. To address gaps in our knowledge about the

## STRENGTHS AND LIMITATIONS OF THIS STUDY

⇒ This is the first realist review of Adolescent-Focused Low-Intensity Life Story Work and it improves our understanding of how this intervention may work for adolescents in out-of-home care.

⇒ The review benefited from ongoing consultations with stakeholders in two separate Content Expert Groups, featuring young adults with care-experience and professionals, as recipients and deliverers of Adolescent-Focused Low-Intensity Life Story Work.

⇒ Content Expert Groups identified gaps in the evidence base, meaning there were limitations in how these aspects could be incorporated into the developing programme theory.

⇒ Future primary research is needed: (1) to further consolidate Adolescent-Focused Low-Intensity Life Story Work components (elements that cannot be changed) and its adaptable periphery (elements that can be changed), (2) build on initial practice guidance offered by this paper, (3) to identify important outcomes for adolescents in care and those who support them and (4) deliver a comprehensive evaluation to enable the therapeutic potential of Adolescent-Focused Low-Intensity LSW to be examined.

impact of Adolescent-Focused Low-Intensity LSW, further primary research is needed to strengthen understandings of how this intervention works (or not) in different contexts.

**PROSPERO registration number** CRD42021279816.

## BACKGROUND

There are currently over 105 000 children and adolescents removed from their homes and looked-after in out-of-home care in for example, foster or residential placements, in the UK.[1–4] Within this population, adolescents living in out-of-home care (henceforth adolescents in care) are the largest group currently in, and the fastest growing age group entering

care in England.[1] They are among the most vulnerable members of society and are six times more likely than the general population to experience mental illness.[5 6] The National Institute for Health and Care Excellence (NICE) describes the need to increase the evidence-base for effective mental health interventions for adolescents in care as an 'urgent research priority'.[7]

Life Story Work (LSW) is an existing transdiagnostic intervention thought to improve the mental health and well-being of those living in care.[8] It aims to improve individuals' sense of identity, relationships with important adults, ethnic heritage, placement stability and mental health, by addressing gaps in self-knowledge, reframing past events and increasing positive future expectations.[7–13] Typical LSW components include a therapeutic alliance (relationship with caring adults), procedures (prompts to action), products (materials/artefacts/life story books), and therapeutic activities (rescripting/reframing).

The intensity of LSW varies along several continua including whether delivery involves support from a specialist professional or not, delivery mode (face-to-face and/or digital), duration (prescribed sessions or ongoing) and intensity of services provided (eg, past and/ or trauma-focused or child led).[14 15] LSW is often delivered in a high-intensity way, involving specialist professionals (eg, Social Workers, Educational Psychologists, Clinical Psychologists, Psychotherapists) input over many sessions and months. This is potentially expensive and impractical, limiting its availability. LSW also tends to be delivered to younger children (aged 0–11 years) meaning adolescents in care (12–18 years old) often miss out.[16]

LSW can be delivered in a lower-intensity way, for example, by foster carers. This increases its accessibility and is sometimes called 'low-intensity' LSW. Low-Intensity Life Story Work (LI-LSW) usually involves a carer (therapeutic alliance) recording (prompts to action) potentially valuable parts of a child's every day present circumstances (product creation). The carer and child discuss these, helping the child to process and consolidate their experiences (therapeutic activities). This in turn serves to generate positive future expectations and helps a therapeutic alliance develop between child and carer.[13 17]

NICE states LSW should be offered to all those living in care.[7] There is currently no accepted standard for LI-LSW delivery or high-quality evidence-based guidance and yet evidence has repeatedly highlighted the harms of poorly undertaken LSW.[18–25] What little guidance there is focuses mainly on younger children in care[20 26–28] The current lack of 'Adolescent-Focused' LI-LSW guidance means adolescents in care experience ongoing inconsistent practices and mental health inequalities leaving them and their carers feeling unsupported.[28] Adolescence is an important period where concepts of self and identity are developing as well as social and emotional practices that are influential to well-being.[29] Therefore a whole-child approach is necessary to ensure adolescents gain the potential benefits of LSW, particularly when delivered taking a low-intensity approach.

To address this evidence and practice gap, we aimed to generate in-depth knowledge to understand if LSW can be delivered to adolescents in care as a low-intensity intervention by asking:

How, why, to what extent, for whom and in what circumstances can LI-LSW interventions, or elements of LSW interventions, be delivered to improve important and relevant outcomes for adolescents with care experience with mental health and well-being needs?

The end goal of the project was to develop a set of initial guidelines for optimising Adolescent-Focused LI-LSW to improve the mental health and well-being and mental health and well-being support offered to adolescents in care.

## METHODS

We conducted a realist review to synthesise evidence to help us understand the importance of context in the delivery of Adolescent-Focused LI-LSW, and the mechanisms by which outcomes, both intended and unintended, are produced. Understanding what works, for whom, in what respects, to what extent, in what contexts is vitally important as adolescents in care are an extremely heterogeneous and complex group interacting with a wide variety of complex services.[28] A realist review was selected because Adolescent-Focused LI-LSW is a complex intervention and how well it works (or not) depends on context, who delivers it and how, with context-sensitive outcomes particularly likely.[8 16] Producing knowledge to understand this complexity will enable commissioners and implementors of Adolescent-Focused LI-LSW to make more informed decisions about if, how, when and where it may be useful.

### Realist review

Realist approaches are grounded in the assumption that the same intervention will not work everywhere and for everyone.[30] A realist review is a theory-driven interpretive type of literature review. Realist reviews are particularly useful for generating knowledge about how complex interventions produce mixed outcomes by generating better understandings of how and why different outcomes occur.[31–33] Central to realist reviews is the creation and iterative refinement of a programme theory. A programme theory describes how an intervention is expected to lead to its outcomes and in which contexts these should occur.[31–33]

Realist reviews begin with an initial programme theory, developed from previous research, lived-experiences and assumptions about how an intervention works. Literature searching is then undertaken, data extracted and synthesised using a realist logic of analysis to develop context–mechanism–outcome configurations (CMOCs). CMOCs are causal explanations describing how and why particular outcomes are generated in particular contexts.[31–33] CMOCs are key components that inform any programme theory as they underpin any causal claims made.

Realist reviews can draw on a breadth of data sources to inform the programme theory. This was important due to the low-quality of the majority LSW literature and the aim of the current review to understand the various contexts within which Adolescent-Focused LI-LSW may occur.[16] Hence, it was important to not limit included documents to solely published academic literature. Conducting a realist review enabled the inherent complexity of the research question to be addressed by using any relevant data that accounted for the different settings and services adolescents in care may experience. Through its focus on commonly occurring causal processes that underlie LSW, the realist approach also potentially provides transferable explanations for how and why other low-intensity mental health intervention strategies 'work' (and do not work) for adolescents in care. This enabled the study to begin to address knowledge gaps highlighted in previous work relating to improving implementation guidance.[8 16]

In this paper, we report on a realist review study conducted between September 2021 and January 2023. A five-step process for conducting the realist review was undertaken following published standards.[32 33] These steps are summarised in table 1 below, with table 2 showing eligibility criteria, with full details about our methods available in our open-access protocol paper.[9]

### Data sources searched

As per our protocol paper,[9] electronic database searches were performed between December 2021 and March 2022. Eight databases were searched: MEDLINE, Embase, PsycINFO, Sociology Collection (ProQuest), CINAHL, CDAS, Web of Science and Social Care Online. To retrieve

**Table 1** Summary of methods

| Step | Aim | Approach |
|---|---|---|
| Step 1: develop an initial programme theory | To identify existing relevant theories that provide explanations of why and how Adolescent-Focused Low-Intensity Life Story Work (AF-LI-LSW) approaches work (or are thought to work), in what contexts they work, to what extent and for whom. | Exploratory searches (CD).<br>Project team knowledge and consultation with stakeholders (all). |
| Step 2: evidence search | To perform a comprehensive literature search to identify data to develop and refine the initial programme theory. | Electronic database searches across multiple databases (see online supplemental file 1).<br>Targeted Google searches to retrieve grey literature produced by local authorities (CD).<br>Citation tracking (EM).<br>Project team knowledge (all).<br>Email alert to capture new literature (CD). |
| Step 3: document selection, data organisation and extraction | To select eligible documents that could include relevant data to contribute to the development and refinement of the programme theory. To describe selected documents and to extract and code relevant data. The full set of extracted data are available on request. | Screen title and abstract and then full text against the inclusion and exclusion criteria (see table 2: inclusion and exclusion criteria (EM and SH).<br>Included documents selected based on the rigour and relevance of documents.<br>10% random sample of documents screened in duplicate with discrepancies resolved through group discussion (EM and SH).<br>Key characteristics of selected documents were extracted into Excel (EM).<br>Full texts uploaded to NVivo and relevant data to the research question and review focus were coded (EM). |
| Step 4: evidence synthesis | To apply a realist logic of analysis to extracted data to develop context–mechanism–outcome configurations (CMOCs) about how AF-LI-LSW improves the mental health support available to adolescents looked-after by social care. CMOCs are central to realist analysis, offering an explanation of the relationships between the conditions an intervention is delivered, the mechanisms by which the intervention works and the outcomes that are produced when delivered within those particular contexts. | The coded data was closely inspected, and interpretations were made about whether evidence was functioning as a context, mechanism or outcome and which CMOC the evidence belonged to (EM and SH).<br>Cross document comparisons were made to build CMOCs, as documents did not always contain evidence that supported the development of CMOCs as evidence for each element of the configuration could not always be found in one document (EM and SH).<br>As the CMOCs were created and refined, judgements were made on how they related to each other in consultation with the project team and where relevant stakeholders (all). |
| Step 5: development of consolidated programme theory and drawing conclusions | To develop a consolidated programme theory containing the CMOCs. | The programme theory was refined iteratively through the development of CMOCs and consultation with stakeholders and project team discussions (all).<br>Judgements were made on the plausibility and coherence of the developing programme theory (all).<br>The consolidated programme theory was written and shared with stakeholders and final amendments were made based on their consultation (all). |

**Table 2** Eligibility criteria

| Inclusion criteria | Exclusion criteria |
|---|---|
| Population: young people who are under the care of a local authority, young people who are 'looked after' or care experienced or adopted young people or their parents/carers. | Research focused solely on parenting style, communicative openness in foster or adoptive families, contact with birth family members. |
| Intervention: Life Story Work, including all activities involving recording, exploring, eliciting accounts of a care experienced person's life or personal history, to have an impact on their understanding of themselves and their identify. And/or Low-intensity interventions that aim to address a mental health or well-being need. | |
| Document type/study design: any. | |
| Other: English language only. | |

relevant PhD theses, we also searched EThoS (British Library) and Proquest Dissertations and Theses. Targeted Google searches were performed to retrieve grey literature produced by local authorities and further documents were obtained through citation tracking and team members' knowledge. The precise full search strategy is shown in online supplemental file 1.

### Patient and public involvement—working with experts via experience

The idea for the project from which this paper reports was driven by the lead author's time in practice and shaped through working with a variety of experts via experience from this conception. This review was informed by the involvement of two experts via experience groups. These groups consisted of a Care-experienced Content Expert Group which involved care leavers (young adults with recent experiences of living in social care), and a professional expert panel (known as the Content Expert Group) which comprised of multidisciplinary professionals with divergent topic area knowledge and/or experiences (ie, Foster Carers, Social Workers, Commissioners, Educational and Clinical Psychologists).

We met with the Care-experienced Content Expert Group and Content Expert Group three times during the project. We asked these groups to review information, bring their own experiences to bear on this, and help us to critique and refine the developing programme theory. These discussions helped clarify important areas that needed explanation (eg, the need for adolescent-focused low-intensity LSW to be flexible) and where the existing literature was lacking (eg, peer-support for adolescents engaging with Adolescent-Focused LI-LSW), thus shaping our analysis. In our final meetings, we asked each group to help us to develop and refine an initial set of guidelines. Both groups also helped develop dissemination strategies, ensuring project outputs were tailored

to different audiences. For example, working with our Care-experineced Content Expert Group we co-designed the following animation to share the core messages of the research to adolescent and the adults that support them https://www.youtube.com/watch?v=geF1WVPkcSY.

## RESULTS

### Document characteristics

In total, 75 documents were included in the review. Documents were published between 1981 and 2022 and were from eight countries (56 from the UK). Included documents were research articles (36%), commentaries or practice guidance (32%), web sources (3%), reports (5%), conference paper (1%), quality standards (1%), books or book chapters (11%) and theses (11%). Of the research articles, 78% were qualitative studies, 4% used mixed methods, 7% were literature reviews, 7% were matched pairs and 4% were randomised trials. Judgements were made concerning the relevance of data within documents and whether data could be used to improve our understanding of how Adolescent-Focused LI-LSW could be implemented. The main reasons for exclusion were documents being about the wrong population or not containing information that contributed relevant data to the CMOCs for Adolescent-Focused LI-LSW. The processes of identifying, screening and selecting documents are illustrated in figure 1: an adapted version of the PRISMA (Preferred Reporting Items for Systematic review and Meta-Analysis) flow diagram. A detailed summary of included documents are available in online supplemental file 2.

### Review findings

The initial programme theory presented a linear explanation of Adolescent-Focused LI-LSW (see figure 2). What was clear within the literature and from consultations with our Care-experienced Content Expert Group and Content Expert Group was the current low-quality of the evidence-base, alongside the need for Adolescent-Focused LI-LSW to be both continual and flexible. Data synthesis resulted in a consolidated programme theory from which a set of initial guidelines for optimising Adolescent-Focused LI-LSW to improve the mental health and well-being and mental health and well-being support offered to adolescents in care was created (see figure 3). Eight initial guidelines are outlined below and also concisely summarised in online supplemental file 3). The narrative summary accompanying each guideline is based on our realist analysis that developed 51 CMOCs, underpinned by the data included in the review. To enhance transparency, we provide a detailed summary of the CMOCs developed and the data underpinning each in online supplemental file 4. We also provide references to illustrate CMOCs to support the report of the results of the review that follows.

### Adolescent-Focused Low-Intensity LSW should be flexible and person-centred (CMOC 1)

Ten documents contributed to the development of this CMOC.[7 20 27 34–40] The review's core finding is the

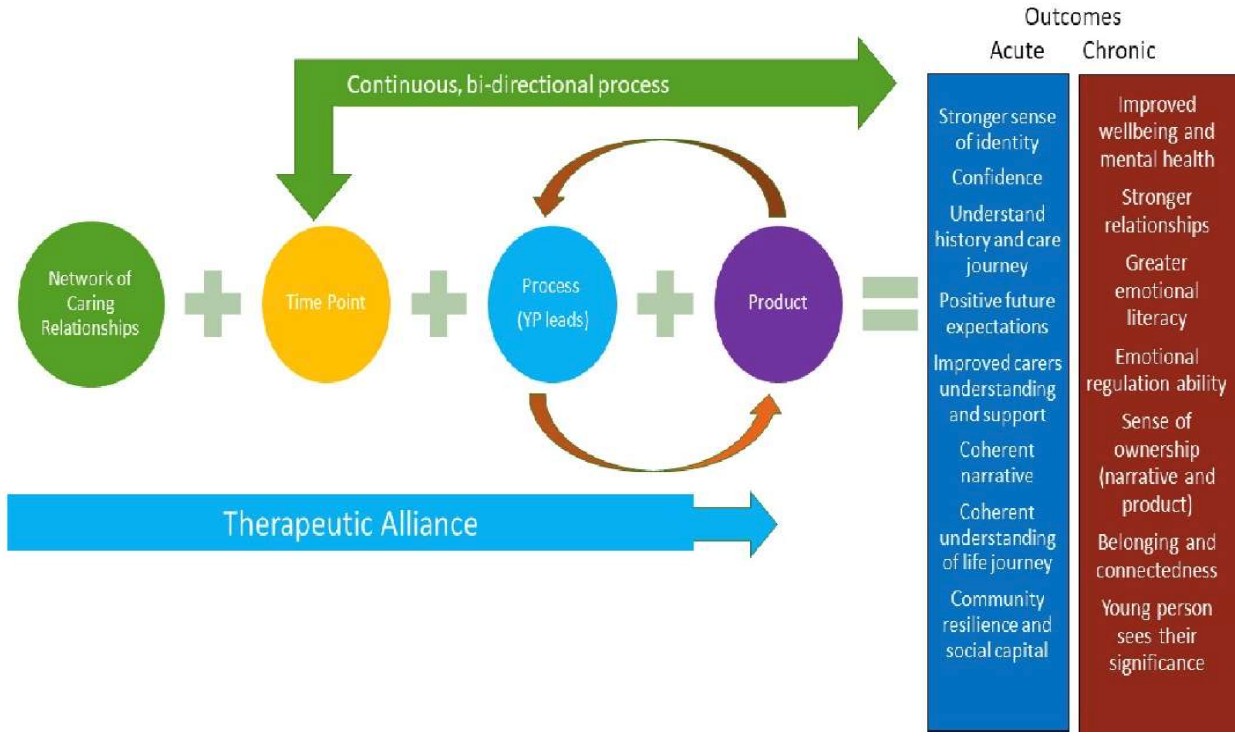

**Figure 1** PRISMA flow diagram providing summary of searching and selection processes. LI, Low-Intensity; LSW, Life Story Work; PRISMA, Preferred Reporting Items for Systematic review and Meta-Analysis.

**Figure 2** Initial programme theory for Adolescent-Focused Low-Intensity Life Story Work.

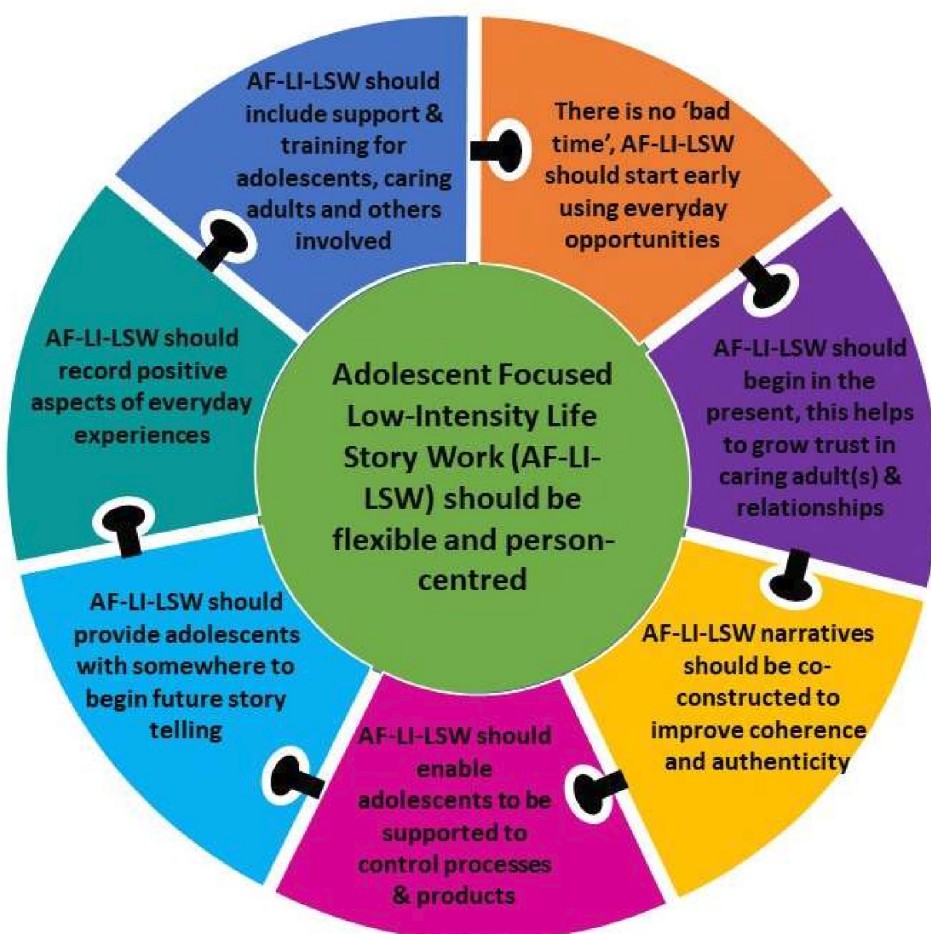

**Figure 3** Consolidated programme theory for Adolescent-Focused Low-Intensity Life Story Work.

importance of Adolescent-Focused LI-LSW being flexible and person-centred. Adapting to the needs, interests and wishes of the adolescent is the key initial guideline that underpins all that follow.

> Take a flexible approach to life story work, and tailor it to the developmental age and needs of the looked-after child or young person.[7]

### There is no 'bad time' to start Adolescent-Focused Low-Intensity LSW so this should start early and make use of everyday opportunities (CMOCs 2–7)

Thirty-one documents contributed to the development of these CMOCs.[7 19–22 27 34–36 38 39 41–59] The review identified two factors associated with when to begin, and how to initiate Adolescent-Focused LI-LSW. It should start immediately and take advantage of everyday opportunities. Beginning early helps preserve information and avoid narrative gaps. This ensures that adolescents have memory prompts available to begin to construct a coherent identity, with these prompts containing everyday events.

> Entries assist in helping the youth preserve memories…. These stories may someday help the youth establish a stronger sense of self based on positive memories of their past.[50]

A caveat within conventional, 'high-intensity' LSW literature is that it should begin at the 'right time'. This 'right time' is often described as when a child is in a stable placement and has trusting relationships to support them as this work involves processing their past, potentially traumatic experiences.[27 34] This is often a barrier to any LSW being implemented. However, an Adolescent-Focused LI-LSW approach focuses on an adolescent's everyday life and future, avoiding an overemphasis on past experiences and freeing carers to begin this work immediately. This relationship may then serve as a vehicle to explore more sensitive previous experiences. However, higher intensity LSW approaches are not the focus of this review and represent a distinct area of need.

### Adolescent-Focused Low-Intensity LSW should begin in the present day as this grows trust in caring adult(s) and relationships (CMOCs 8–10, 12–18 and 30)

Forty-three documents contributed to the development of these CMOCs.[7 19–21 27 34 37 38 40 41 46–49 52–55 59–81] Difficulties in establishing and maintaining trusting relationships was a main theme. Adolescent-Focused LI-LSW should provide a platform to engage in positive activities between caring adults and adolescents, encouraging one-to-one time together and increasing

communication.[7 19 27 34 37–39 47 60–64] These shared activities should encourage increased communication between the adolescent and caring adults, strengthening relationships and helping to grow trust and a therapeutic alliance between the adolescent and caring adult(s). This helps the adolescent to understand themselves better as well as helping the caring adult become better attuned to the adolescent's needs, improving their ability to care for them effectively.

> Young people seemed to trust relationships where others would sensitively listen to their confidences, whilst remaining unconditionally accepting of them as a person….[63]

As well as contributing to interpersonal outcomes such as increased connection, communication, sense of belonging, enhanced ability to understand others behaviour and placement stability, the literature indicated the potential value of Adolescent-Focused LI-LSW has on intrapersonal outcomes such as increased self-worth and emotion regulation. Adolescents can develop their emotional regulation skills through discussing thoughts and feelings and having a caring adult modelling adaptive behaviours to challenging experiences.

> If adults respond to children's distress in a calm but engaged way they demonstrate an alternative way of managing stress.[82]

### Adolescent-Focused Low-Intensity LSW should involve co-construction of narratives because this improves coherence and authenticity (CMOC 23–29 and 31)

Forty-four documents contributed to the development of these CMOCs.[18–22 26 27 34–39 41 42 45 46 48 49 52–55 57 59 63 65 68 69 71 74 77–79 83–92] The review shows narratives most valued by adolescents were constructed collaboratively with caring adult support. The therapeutic alliance enabled adolescents to feel supported to see a range of perspectives when interpreting events. When adolescents are actively involved in the co-construction of narratives, these narratives become more coherent and feel authentic because they reflect adolescents' lived experiences.

> The therapeutic process involves creating opportunities to open up conversational spaces. These spaces provide young people with opportunities to share accounts of their own lives, in their own words.[48]

Literature indicates that adolescents in care may construct self-limiting accounts of their lived experiences due to their chronic and/or traumatic nature.[39 41 49] Evidence indicates that poorly executed LSW has negative impacts.[18–22] Having caring adult support can help adolescents reconstruct narratives, reducing self-blame and self-limiting interpretations of events and circumstances. When alternatively framed narratives are available, adolescents can become increasingly aware of their positive attributes. This can increase self-esteem and the availability of positive future expectations.

> There is something powerful in how people come to see their stories and make sense of their experiences. This is the heart of resilience – the ability of a person to understand their story in such a way that it creates opportunity rather than limits it.[79]

The inclusion of multiple voices and perspectives increases adolescent's awareness that there are different ways to interpret situations and circumstances. Relevant literature was limited in this area, perhaps pointing towards its inherent complexity and clashes between adolescent agency and professionalised notions of 'accuracy'.

> Granting children control over narratives aids identity formation but left unchallenged their perspective of events may also become one-sided and inaccurate.[39]

However, both of our stakeholder groups indicated the value of including significant individuals, including birth relatives and siblings in Adolescent-Focused LI-LSW. Literature indicated that when/if significant individuals are positively involved in Adolescent-Focused LI-LSW activities, this can facilitate the establishment of stronger relationships because of the interactions that result from the activity. The involvement of significant individuals may also increase adolescents' sense of identity because they are able to access and process information about their heritage.

### Adolescents should be supported to control how their lives are recorded and preserved in Adolescent-Focused Low-Intensity LSW whenever and wherever possible (CMOC 32–42)

Adolescents should control how their everyday life experiences are reflected on and preserved including the processes used for recording and preserving lived experiences as well as the resulting products.

### Processes for recording and preserving (CMOC 32–36)

Twenty-six documents contributed to the development of these CMOCs.[7 18 20 22 27 34–37 39 40 45–48 51 53 55 62 64 65 78 93 94] The review indicates that being able to exert control over the process increases adolescents' sense of ownership. This helps adolescents engage better and feel their opinions are valued. This is invaluable for adolescents in care as often due to their circumstances their sense of power can be reduced.

> Children can have ownership of their story work via choosing which objects to story/not story and by dictating the pace at which the work progresses… (Fostering Social Worker, Ben)[39]

There was limited literature surrounding how, when and who should be responsible for making LSW accessible for adolescents with special educational needs. The available literature and our stakeholder groups indicated the importance of the adolescents' support network feeling equipped with the knowledge and skill to make adaptions where necessary. Caring adult(s) will benefit from, for example, training materials developed by speech and

language therapists. This may include written and visual resources that increase their understanding of how to make adaptations to improve adolescent's abilities to engage with Adolescent-Focused LI-LSW in a meaningful way. Though research was sparse in this area, our Care-experienced Content Expert Group and Content Expert Group stated this was important to ensure that Adolescent-Focused LI-LSW is flexible and person-centred.

### Products for recording and preserving (eg, artefacts) (CMOCs 37–42)

Thirty-one documents contributed to the development of these CMOCs.[18 20–22 26 27 34 35 38–42 45 46 48–50 52–55 78 85 90 93–97] The literature highlighted that when an adolescent has editorial power to control what is recorded, they can feel more positive about the product(s). This is because what is recorded aligns more closely with their lived experiences. The risk of LSW products being lost or damaged was a theme within the literature. Literature and our Care-experienced Content Expert Group and Content Expert Group repeatedly illustrated the benefits of digital tools/platforms to create and store editable products. These products were cited as holding meaning over time because they held identity information about adolescents, their relationships and spaces at a point in time and could be updated and edited when needed.

> Digital tools offer the ability to help young people to express thoughts and feelings which can be continually updated, edited and reflected upon. Using interactive computer-based mediums in this way grants the user flexibility to make changes frequently and easily.[45]

The importance of both physical, audio and visual artefacts was emphasised. Physical objects can increase a sense of connection to others and/or past experiences via the tangible nature of the artefact that was present during life experiences. Audio and visual formats, including capturing the actual voice or video recording of significant individuals can be powerful, helping adolescents to situate memories and feel connected to their experiences and significant individuals.

> For looked after and adopted children, physical objects are often the only remaining link to their past; a portal to stories of birth families, former homes, and significant people.[39]

### Adolescent-Focused Low-Intensity LSW should provide adolescents with somewhere to begin future telling (CMOC 11 and 43)

Eight documents contributed to the development of CMOCs.[34 35 38 39 41 49 62 75] Our stakeholder groups repeatedly emphasised the life course nature of care-experience and enduring impacts that missing information has on identity and self-esteem throughout one's life. During adolescents' time in care, they can use their collection of preserved memories as storytelling prompts (eg, a photo with a date, time, location, names of people and activity relating to) with caring adults, which helps others to understand them, strengthening relationships. This may include transitions within and beyond care. When preserved, life experiences can become a bank of narrative starting points (artefacts from which to begin reminiscing) for adolescents to share stories about themselves with future audiences.

> The book acted as an aide to him telling me about his life, the things and people that are important to him.[49]

### Adolescent-Focused Low-Intensity LSW should record positive aspects of everyday experiences (CMOC 44–46)

Twenty-one documents contributed to the development of these CMOCs.[19 20 26 27 34 37 38 47 49 52 54 59 66 69 80 81 86 95 98 99] The review highlighted the value of using Adolescent-Focused LI-LSW to capture adolescents' 'everyday magic' (ie, an individual's mannerisms, characteristics and their idiosyncrasies, humour and histories of relationships shared with those around them and everyday successes) helping adolescents to feel more positive about themselves and their relationships.[38 39 47 49 52 66 80 81 86 98] When achievements are preserved and reflected adolescent's ability to develop positive identities and future expectations are supported because the availability of positive self-narratives for interpreting their experiences are more readily available. They can become better at identifying future achievements themselves.

> …it can help the child to thicken positive counter-narratives impacting positively on the dominant stories the child has and is able to tell about themselves, particularly thickening stories around their strengths, worth and belonging.[49]

### Adolescent-Focused Low-Intensity LSW should include support for adolescents, caring adults and others involved (48–51)

Twenty-five documents contributed to the development of these CMOCs.[7 27 34 37–39 41 48 49 51 54 55 58–60 66 68 70 72 77 79 82 87 95 100] As mentioned above, Adolescent-Focused LI-LSW should be flexible and person-centred. The literature highlights the essential role confidence has on the ability of caring adults to support Adolescent-Focused LI-LSW. It is important that caring adult(s) understand the adolescents' individuality and cultural heritage to develop their awareness of the potential significance of everyday lived experiences to promote appropriate preservation and reflection. Caring adults' need to be supported via training and ongoing supervision to improve their confidence and skills to support and facilitate Adolescent-Focused LI-LSW.

> Training to ensure a consistent approach to life story work could be incorporated into existing training.[7]

The literature indicated that training is most impactful when accompanied by supervision to consolidate

learning. Supervision for caring adults ensures advice and guidance is readily available as well as providing a self-reflective space to process emotions and improve practice.

> Recognition of children's entitlement to a coherent narrative needs to be embedded in practice at the micro level. This can only occur if appropriate supports are in place at the organisational level, facilitated by macro-level priority….[41]

Alongside providing training and supervision for caring adults, the potential role of peer-support, for adolescents and caring adults and when implementing Adolescent-Focused LI-LSW was repeatedly mentioned by our expert groups. Adolescents may benefit from peers supporting their understanding of Adolescent-Focused LI-LSW as during adolescence it is widely acknowledged that peers become increasingly influential,[101] meaning there is the potential for peer support to be impactful. Peer-support for caring adults also hinted at ways to develop communities of practice. Literature was extremely limited on the role of peer-support for both groups. However, other models of practice have employed peer-support models with these populations with promising outcomes (eg, The Mockingbird programme). This is an important direction for future research.

## DISCUSSION
### Summary
Adolescent-Focused LI-LSW involves an adolescent and/or caring adult(s) recording potentially valuable parts of an adolescent's everyday life (eg, diary entry or picture). The caring adult(s) and adolescent discuss these, helping the adolescent to process and consolidate their experiences. This in turn serves to generate positive future expectations and helps a therapeutic alliance develop between adolescent and caring adult(s).[7 27 37 38 47 49 61–63 84–86 88]

### Comparison with existing literature
In a recent systematic review targeting the mental health and well-being of care-experienced children and young people,[102] only one paper related to LSW was included.[66] The authors of the systematic review described the paper as providing a limited description of intervention implementation and acceptability.[102] This dearth of good quality evidence on LSW as a whole is contrary to repeated high-quality evidence illustrating that key components that make up LSW are effective. For example, first we know that being able to express emotions, change negative cognitions, recall positive life experiences, create positive future expectations and therapeutic alliances are key targets to promote mental health, especially for vulnerable adolescents.[13 17 28 103–105] Second, we know that trusting relationships with key adult(s) and peer(s) are frequently indicated as important in recovering from traumatic life experiences beyond childhood.[106 107] Third, we know that positive psychological and reminiscence interventions are

impactful.[11] Finally, we know that better access to high-quality mental health and well-being support can build self-esteem and belonging, themes frequently cited in reports about the mental health and well-being needs of adolescents in care[28 102 103 108]

Systematic review evidence examining LSW interventions for children and adolescents in care concluded that LSW had potentially far-reaching benefits but an 'immature' evidence-base.[12 109] However, the existing evidence does indicate that the lack of 'robust implementation guidance, efficacy or cost-effectiveness studies'[16] is problematic and that 'greater implementation knowledge' is needed to examine therapeutic outcomes.[12] A recent attempt to improve this evidence-base was inconclusive,[8] with this rare example of a randomised controlled trial in this area ultimately lacking statistical power. The authors reflect on a number of design issues including 71% of the young people in the active arm not receiving the intervention, a lack of carer engagement due to restrictive intervention processes and issues with the extraction of the selected primary outcome measure from social care records.[8] However, the process evaluation did indicate widespread positive qualitative feedback with young people reporting feeling more connected to carers. Echoing earlier systematic review findings, authors concluded that any future trial must be sufficiently powered and sensitive to how context shapes intervention delivery.[8]

### Limitations
This review has several limitations. The literature that informed the CMOCs were supported by relevant and best currently available evidence. However, when framed by the hierarchy of evidence for therapeutic studies[110 111] most of the available evidence is 'low-quality'.[8 28 112–114] Existing literature was mainly qualitative in nature, tended to focus on LSW undertaken with younger children (0–11 years old) and higher intensity delivery models. While the available evidence with an adolescent focus was invaluable for illustrating 'good' and 'bad' practices, it did not provide enough detailed information about implementation or how different circumstances, including different placement contexts impact on outcomes or how Adolescent-Focused LI-LSW operated over time. This is problematic from a realist perspective since the same body of qualitative evidence is used to inform the Contexts, the Mechanism and the Outcomes meaning the review teeters on the edge of working in one data medium method rather than being multimethod.[115] This means that we were unable to effectively test the programme theory and, in particular, the potential of contextual impacts on intervention effectiveness. This is a substantial gap in our understandings of pre-existing practice. It must be urgently addressed as understanding what works, for whom, in what respects, to what extent and in what contexts is vitally important as adolescents in care are an extremely heterogeneous group interacting with a wide variety of complex services.[28 102] How context influences mechanisms and outcomes is therefore vital.

The current review adds to this by improving our understanding of how, why, what and whom should be involved in the delivery of Adolescent-Focused LI-LSW to produce potentially therapeutic outcomes. Our realist review, the first of its kind in this area, created a programme theory for Adolescent-Focused LI-LSW and used this to produce initial guidance for how current practice should be improved. Our review is unique in describing considerations that support adolescents in care to experience ownership over how, when, why and in what contexts their everyday lived experiences are narrated, recorded and reflected on during their time living in social care.

A flexible and person-centred approach is key, both in generating feelings of investment and empowerment but also in implementing Adolescent-Focused LI-LSW. An important task for caring adults is identifying and empathising with adolescents, particularly in relation to their desired level of involvement within Adolescent-Focused LI-LSW, which may fluctuate over time and across contexts. This review illustrates that Adolescent-Focused LI-LSW is to likely have core components (elements that cannot be changed) and is likely to have an 'adaptable periphery' (elements that can be changed) capable of adjusting to contextual factors.[116] More research is needed to identify these components to allow learning to be transferable between contexts.

### Implications for practice

Adolescent-Focused LI-LSW is not a magical solution to improving the mental health and well-being outcomes for adolescents in care. However, considering NICE states LSW should be offered to all those living in care,[7] that it is endorsed by legislation underpinning its usage[117–120] and widely valued by those living and working within social care,[10 18–22 27 34 81] we hope our initial practice guidance (summarised in online supplemental file 3) can begin to improve existing practices. In implementing this guidance, there is a need to consider where, how and in what format Adolescent-Focused LI-LSW products are stored, who has ownership of this content and how it can be transferred when those living in care move within and beyond the care system. Our Care-experienced Content Expert Group and Content Expert Group both repeatedly highlighted the potential of digital platforms/devices. They felt that a digital platform had the potential to be effectively used within Adolescent-Focused LI-LSW to record and collate everyday life experiences. This is something that members of the team have written about before.[35 45 46 121] However, existing research was sparse in this area and research to understand the value of digital platforms to support LSW, such as CaringLife, is ongoing at the time of publication.

### Implications for research

Given the complexity of Adolescent-Focused LI-LSW it is perhaps unsurprising that many studies reported variation in interventions and their implementation. However, there were also issues within many studies where the description of both were either absent or underdeveloped. This can be remedied by researchers using tools such as the Template for Intervention Description and Replication (TIDieR) or Standards for Reporting Implementation Studies (StaRI) checklists to enable authors to efficiently outline interventions and report their implementation in detail.[122 123] Furthermore, to develop CMOCs or some aspects of CMOCs which were only supported by limited evidence, we drew on expert opinion. These CMOCs were discussed with our expert groups who felt they were important to include. Gaps in these areas, (eg, use of digital media, peer-to-peer support, engaging with adolescents in care with Special Educational Needs and involving significant individuals (including birth relatives), effectiveness and implementation of training and supervision) also represent important future research directions.

Further research is required to strengthen our understanding of how Adolescent-Focused LI-LSW delivery is impacted by individual circumstances experienced by adolescents living within different placement contexts (eg, foster, residential, kinship), as well as what therapeutic outcomes are important for adolescents, caring adults and social care more broadly before any future evaluations may be undertaken. Future primary research is therefore needed: (1) to further consolidate Adolescent-Focused LI-LSW core components (elements that cannot be changed) and its adaptable periphery (elements that can be changed) to enable transferability, accessibility and implementation, (2) to identify important and relevant outcomes for adolescents in care and those who support them, (3) use this knowledge to improve the eight initial guidelines offered by this study concisely summarised in online supplemental file 3 and create actionable recommendations to improve the quality and consistency of Adolescent-Focused LI-LSW; (4) use this knowledge to inform the design of a comprehensive, appropriately powered evaluation, including economic and process elements of Adolescent-Focused LI-LSW which would begin to address the current scant evidence-base. When considering Adolescent-Focused LI-LSW as an intervention, the evaluation of outcomes may be challenging. As a result, evaluations measuring the impact of Adolescent-Focused LI-LSW may need to focus on intermediate outcomes, for example, relationship with caring adult and longer-term outcomes, such as self-esteem, identity coherence and mental health outcomes more broadly over an extended follow-up time.

### CONCLUSION

To our knowledge, this is the first realist review exploring how, why, to what extent and for whom Adolescent-Focused LI-LSW may work (or not) for adolescents in care. Our programme theory and initial guidelines provide important ways to improve the nature of good practice when delivering Adolescent-Focused LI-LSW to adolescents in care and produce benefits for them, their carers and health/social-care professionals.

However, to make this approach accessible to all adolescents in care a programme of research is needed to clarify how contexts impact on outcomes to enable greater

transferability and more successful implementation. Additionally, to improve the current low quality evidence base, a fully powered evaluation with embedded economic and process evaluations and long-term follow-up is required to generate knowledge regarding the therapeutic potential of this highly valued yet inconsistently implemented mental health intervention. Importantly, if not done properly Adolescent-Focused LI-LSW, like other interventions could do harm.[124] Hence, any future evaluation is not just about optimising delivery but also minimising harm. Given that research consistently demonstrates that adolescents in care are among the most vulnerable in society, this difficult but not impossible challenge needs urgent action.

**Author affiliations**
¹School of Education and Lifelong Learning, University of East Anglia, Norwich, UK
²Norfolk and Suffolk NHS Foundation Trust, Norwich, UK
³Nuffield Department of Primary Care Health Sciences, University of Oxford, Oxford, UK
⁴Division of Psychology and Language Sciences, University College London, London, UK
⁵School of Social Work, University of East Anglia, Norwich, UK
⁶National House Project, Crewe, UK
⁷The Fostering Network, London, UK

**Contributors** SH: Conceptualisation, Methodology, Writing—Original draft preparation, Writing—Review and Editing, Supervision, Funding acquisition, Project administration, Guarantor. EM: Original draft preparation, Conceptualisation, Methodology, Funding acquisition, Writing—Review and Editing, Project administration, Visualisation, Project administration, Investigation. CD: Funding acquisition, Conceptualisation, Formal analysis, Data curation, Original draft preparation, Writing—Review and Editing, Visualisation, Investigation. RH: Funding acquisition, Conceptualisation, Writing—Review and Editing. EN: Funding acquisition, Conceptualisation, Writing—Review and Editing. RB: Methodology, Conceptualisation, Writing—Review and Editing, Resources. KW: Funding acquisition, Conceptualisation, Writing—Review and Editing, Resources. JW: Funding acquisition, Conceptualisation, Writing—Review and Editing. GW: Funding acquisition, Conceptualisation, Methodology, Original draft preparation, Writing—Review and Editing, Supervision.

**Funding** This project is funded by the National Institute for Health Research (NIHR) for Patient Benefit Programme (Grant Reference Number NIHR 201963).

**Competing interests** SH offers consultancy services in social media and Digital Life Story Work via www.digitallifestorywork.co.uk. KW is the CEO of The Fostering Network. The remaining authors declare that they have no known competing financial interests or personal relationships that could have appeared to influence the work reported in this paper.

**Patient and public involvement** Patients and/or the public were involved in the design, or conduct, or reporting, or dissemination plans of this research. Refer to the Methods section for further details.

**Patient consent for publication** Not applicable.

**Ethics approval** Not applicable.

**Provenance and peer review** Not commissioned; externally peer reviewed.

**Data availability statement** All data relevant to the study are included in the article or uploaded as supplementary information.

**ORCID iDs**
Simon P Hammond http://orcid.org/0000-0002-0473-3610
Claire Duddy http://orcid.org/0000-0002-7083-6589

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
