## [Reviewer comments · BMJ Open]

ARTICLE DETAILS

TITLE (PROVISIONAL)	Improving the mental health and mental health support available to adolescents in out-of-home care via adolescent-focused low-intensity life story work: A realist review
AUTHORS	Hammond, Simon; Mickleburgh, Ella; Duddy, Claire; Hiller, Rachel; Neil, Elsbeth; Blackett, Rosie; Williams, Kevin; Wilson, Jon; Wong, Geoff

VERSION 1 – REVIEW

REVIEWER	Tuomainen, Helena University of Warwick, Warwick Medical School
REVIEW RETURNED	11-Jun-2023

GENERAL COMMENTS	This review focuses on an intervention, low intensity life story work, that can support the mental health and wellbeing of young people in care, thus is of great importance. The realist review has been conducted well and the amount of literature that has been reviewed to develop CMOCs is impressive. However, as the authors state, the quality of the evidence in this area is low. Nevertheless, by using an expert group to support the review and analysis, the authors have been able to generate 51 CMOCs that have informed the development of the consolidated programme theory and guidance. Considering this commendable work, the discussion seems truncated, especially section on comparison with existing literature. It would be good to be reminded of the core components and the elements that can be changed. Further reflection of findings would be beneficial. Were the results expected? Could these have been developed without the realist review? Comparison of the process and findings in relation to other realist reviews that have been conducted to improve understanding of interventions and develop guidelines would be possible here. In addition, the limitations section could include strengths of the study. Whilst the authors have published a protocol paper with detailed methods, and included a summary of these in Table 1, I would expect to find key information on methods in the text, such as eligibility criteria, especially as this is included in the abstract. The terminology regarding ‘adolescents in care’ is very varied in the manuscript. It would be better to stick to one or two ways of describing them. The authors have used several abbreviations/acronyms; I would prefer these to be written in full, even in the abstract, to improve legibility. LSW is fine. Also, be consistent with capital letters
--

	(Programme Theory or programme theory or Initial Programme Theory). The documents in Supplementary file 3 could be organised alphabetically based on the type of publication. This would help recognition of the type of evidence used for the review. The column with study design/methods includes information regarding type of publication. Either add 'type of publication' in this column heading or add extra column to contain this information. In Supplementary file 2, explain abbreviations, and ideally add reference numbers as this information is in text. Further minor comments: Abstract: In objectives, explain high-intensity if wordcount permits. In eligibility criteria, improve term for children and adolescents in care. In results, improve sentence starting L40 'The analysis found...' Strengths and limitations: Improve first sentence; 'invaluable' seems superfluous in last point. Introduction and background (choose one or the other): L32 Use full term for NICE. Suggest improving background structure by adding information from the beginning of fourth paragraph after this sentence (L42-45). Then continue with explanation regarding high-intensity and low-intensity versions of LSW. In fifth paragraph, suggest writing 'inconsistent practices' before 'mental health inequalities'. In last paragraph you write about 'initial' guidelines. This generates expectation of subsequent guidelines. You might like to address this. Methods: 'To address this gap' is superfluous. L29 'They are' is not correct here. L44, a reference is missing. Past and present tense is mixed in third paragraph under 'Realist review'. Patient and public involvement: The first sentence could be worded better. 'Once funded' is superfluous. Document characteristics: L46 cut 'include' Review findings: These have been clearly presented or summarised. Acronyms are disturbing. p. 7 (or 8 of 78) L45. Sentence starting 'However, as we were interested...' seems superfluous. p. 9 (10 of 78) L24 (CMOC 32 – 36 – not 26); L50 suggest naming sub-title as 'Products for preserving LSW' In Supplement CMOCs 43 and 44 are missing 'M' p. 11 L31-2 cut second 'repeatedly'; L36 add 'in' or 'for' Discussion: Comparison with existing literature: Consider moving last sentence regarding research to 'Implications for research' Implications for practice: LSW is not a silver bullet to what? Not everyone might be familiar with this expression. The last initial practice guidance in supplementary file 4 is unclear.
--	---

REVIEWER	Banwell, Emily
	The University of Manchester, Manchester Institute of Education
REVIEW RETURNED	21-Jun-2023

GENERAL COMMENTS

Abstract:

A strong abstract.

1. In the objectives section, you mention that LSW is typically delivered in a high-intensity (HI) way, before mentioning that your work focusses on low-intensity (LI), for the reasons you state, that there is little available guidance, and a lot of variation in approach. A quick sentence/few words on why your work on LI approaches is important (e.g., you mentioned in the intro that HI is expensive and less accessible) would really enhance your objective.
2. The conclusion part should tie in more closely with your results. You say that further research is needed, which is valid, but you haven't provided any indication, here, of what yours contributes to the field. How do you know that more research is needed? RE word count, I would suggest that words could be removed from the "analysis" section of the abstract, as it is currently very "jargon heavy" for an abstract.

Introduction:

1. I'm not sure how well "looked after by social care" or "living in care" translates to other countries, so it would be worth expanding this in the opening paragraph, for example that these children are/have been in the care of their local authority, often in foster care etc.
2. You give a good description of what low intensity LSW entails, but providing details of what HI-LSW looks like by comparison would make the distinction clearer, both in terms of the therapeutic process, and how it differs in the time spent on it... basically, what are the key features that differentiate HI and LI, and how do these differences impact outcomes? I know that this difference isn't a key feature of your work, but a few lines on this would really highlight the importance of good research on LI-LSW.
3. Along similar lines, you mention that older children/adolescents often miss out, both in guidance and practice. But what makes LSW so important for adolescents? I imagine this will be to do with the level of insight and understanding that they have compared to younger children, and that the onset of most mental health conditions is in the teenage years (you do mention that the lack of guidance might influence the mental health inequalities, so this might be an appropriate juncture to add this), hence why a whole-childhood approach is necessitated... again, this will develop your rationale, making it abundantly clear to the reader why you are focussing on adolescents, that goes beyond "there isn't much for them".

Methods:

1. Another quick point relating to rationale. Your first paragraph in this section is a little underdeveloped. Some unpacking of the contextual complexity of application of AF-LI-LSW, beyond adding a couple of citations, would help. What contexts has the literature explored? Under what circumstances might it work or not? What elements of this have not been explored, hence why you are conducting your study?
2. A well-selected and well-explained methodology. I had never heard of this type of review before, but it's definitely been added to my mental list, thank you for that!
3. Please include, somewhere, an example search strategy (including terms), and how you devised these. Apologies if I missed this somewhere.
4. You haven't provided any ethical procedures concerning the expert group. This is especially important given that some of them

	were vulnerable young people. Please describe the ethical approval you got, or else, how you decided this wasn't needed. Please also briefly describe the procedures of recruitment and consent that you undertook. Results:  1. What criteria did you use to conclude that the evidence base was low quality? 2. Overall, though, a detailed results section, where links with literature and your groups were clearly made. Nicely done. Discussion:  1. "Existing literature was mainly qualitative in nature". This isn't a valid limitation without further explanation. Quite the contrary, at first glance I can only see how qualitative work is an asset within the field you're looking at (your study focusses on nuance and complexity, which can't always be gauged from statistics). Whilst this may be my ignorant view, you nevertheless need to say why this is a limitation, i.e., how would the presence of more quantitative studies have mitigated this limitation? What else would such studies have provided your review? 2. "Silver bullet" – avoid using metaphors or sayings, as they aren't necessarily understood by people for whom English isn't their first language. Perhaps change to "magical solution" or "simple fix"? 3. I think your summary in supplementary file 4 is worth referring to at another juncture, perhaps at the beginning of the results section. This is a really nice condensed summary that someone might prefer to read if they are just skimming through your paper, so signposting it there would be useful.
--	---

VERSION 1 – AUTHOR RESPONSE

Reviewer 1:

Dr. Helena Tuomainen, University of Warwick

Comments to the Author:

This review focuses on an intervention, low intensity life story work, that can support the mental health and wellbeing of young people in care, thus is of great importance. The realist review has been conducted well and the amount of literature that has been reviewed to develop CMOCs is impressive. However, as the authors state, the quality of the evidence in this area is low. Nevertheless, by using an expert group to support the review and analysis, the authors have been able to generate 51 CMOCs that have informed the development of the consolidated programme theory and guidance.

Reviewer 1: Considering this commendable work, the discussion seems truncated, especially section on comparison with existing literature. It would be good to be reminded of the core components and the elements that can be changed. Further reflection of findings would be beneficial. Were the results expected? Could these have been developed without the realist review? Comparison of the process and findings in relation to other realist reviews that have been conducted to improve understanding of interventions and develop guidelines would be possible here. In addition, the limitations section could include strengths of the study.

- Author's response – We thank the reviewer for their suggestions and have heavily revised and expanded these sections to address these points.

Reviewer 1: Whilst the authors have published a protocol paper with detailed methods, and included a summary of these in Table 1, I would expect to find key information on methods in the text, such as eligibility criteria, especially as this is included in the abstract.

- Author's response – We thank this reviewer for their comment, we were aware of word count and repetition given the protocol paper. Hence, in line with previous publications using similar methods to our own in this publication (e.g. Duddy, Gadsby, Hibberd, Krska & Wong, 2022) we chose to summarise these in table one. We have however, included eligibility criteria as requested in Table 2.

Reviewer 1: The terminology regarding 'adolescents in care' is very varied in the manuscript. It would be better to stick to one or two ways of describing them.

- Author's response – Thanks to the reviewer you for highlighting this. The varying of terms has now been changed so that the term "adolescents in care" is used throughout the manuscript. We have tried to ensure this label is fully unpacked in the opening sections of the paper.

Reviewer 1: The authors have used several abbreviations/acronyms; I would prefer these to be written in full, even in the abstract, to improve legibility. LSW is fine. Also, be consistent with capital letters (Programme Theory or programme theory or Initial Programme Theory).

- Author's response – Many thanks to the reviewer for their comment. The use of abbreviations and acronyms has now been edited. This includes using the full name for the approach, "Adolescent-Focused Low-Intensity LSW" instead of AF-LI-LSW. The full names for the Care-experienced Content Expert Group and Content Expert Group have been used throughout instead of using CCEG and CEG. Instead of using abbreviations the phrases "programme theory" and initial programme theory" have been used throughout.

Reviewer 1: The documents in Supplementary file 3 could be organised alphabetically based on the type of publication. This would help recognition of the type of evidence used for the review. The column with study design/methods includes information regarding type of publication. Either add 'type of publication' in this column heading or add extra column to contain this information.

- Author's response – We thank the reviewer for this suggestion. However, in keeping with other realist reviews published in this journal, we wish to keep the table order in alphabetical order by authors (please see example, https://bmjopen.bmj.com/content/bmjopen/suppl/2022/04/05/bmjopen-2021-057009.DC1/bmjopen-2021-057009supp004_data_supplement.pdf). We also feel that the section 'Document characteristics' assists recognition of the type of evidence we include which may also help reader engage with evidence types included.

Reviewer 1: In Supplementary file 2, explain abbreviations, and ideally add reference numbers as this information is in text.

- Author's response – We have made these amendments and thank the reviewer for their insight.

Reviewer 1: Further minor comments: Abstract: In objectives, explain high-intensity if wordcount permits. In eligibility criteria, improve term for children and adolescents in care. In results, improve sentence starting L40 'The analysis found...' Strengths and limitations: Improve first sentence; 'invaluable' seems superfluous in last point.

- Author's response – Thank you for highlighting these areas. Where word count permitted, all of the suggested changes have now been made to the abstract.

Reviewer 1: Introduction and background (choose one or the other): L32 Use full term for NICE. Suggest improving background structure by adding information from the beginning of fourth paragraph after this sentence (L42-45). Then continue with explanation regarding high-intensity and low-intensity versions of LSW. In fifth paragraph, suggest writing 'inconsistent practices' before 'mental health inequalities'. In last paragraph you write about 'initial' guidelines. This generates expectation of subsequent guidelines.

- Author's response – We thank the reviewer for their helpful comments.
- It has now been changed to just using "Background" rather than "Introduction and Background."
- The full term for NICE has now been used.
- The suggested changes to the structure of the explanation of high and low intensity versions of LSW has been actioned.
- This is the first output from a ongoing programme of research, hence as we conclude, primary research is needed to develop more actionable recommendations. This is why we use the term initial guidelines.

Reviewer 1: Methods: 'To address this gap' is superfluous. L29 'They are' is not correct here. L44, a reference is missing. Past and present tense is mixed in third paragraph under 'Realist review'.

- Author's response – We thank the reviewer their helpful comments and have actioned accordingly.

Patient and public involvement: The first sentence could be worded better. 'Once funded' is superfluous. Document characteristics: L46 cut 'include'

- We thank the reviewer their helpful comments and have actioned accordingly.

Reviewer 1: Review findings: These have been clearly presented or summarised. Acronyms are disturbing.

p. 7 (or 8 of 78) L45. Sentence starting 'However, as we were interested...' seems superfluous

p. 9 (10 of 78) – issue with platform not paper upload

L24 (CMOC 32 – 36 – not 26);

L50 suggest naming sub-title as 'Products for preserving LSW

In Supplement CMOCs 43 and 44 are missing 'M'

p. 11 L31-2 cut second 'repeatedly' - L36 add 'in' or 'for'

Discussion: Comparison with existing literature: Consider moving last sentence regarding research to 'Implications for research'

- Author's response – We thank the reviewer their helpful comments and have actioned accordingly.

Reviewer 1: Implications for practice: LSW is not a silver bullet to what? Not everyone might be familiar with this expression.

- Author's response – We thank the reviewer their suggestions and have made this clearer both in terms of expression but also topic area.

Reviewer 1: The last initial practice guidance in supplementary file 4 is unclear.

- Author's response – We have edited this to make this clearer.

Reviewer: 2

Dr. Emily Banwell, The University of Manchester

Comments to the Author:

Abstract:

A strong abstract.

1. In the objectives section, you mention that LSW is typically delivered in a high-intensity (HI) way, before mentioning that your work focusses on low-intensity (LI), for the reasons you state, that there is little available guidance, and a lot of variation in approach. A quick sentence/few words on why your work on LI approaches is important (e.g., you mentioned in the intro that HI is expensive and less accessible) would really enhance your objective.

- Author's response – We thank the reviewer for their helpful comments. An additional sentence has been included within the objectives to highlight that low-intensity approaches can be more accessible and less expensive

2. The conclusion part should tie in more closely with your results. You say that further research is needed, which is valid, but you haven't provided any indication, here, of what yours contributes to the field. How do you know that more research is needed? RE word count, I would suggest that words could be removed from the "analysis" section of the abstract, as it is currently very "jargon heavy" for an abstract.

- Author's response - Thanks to the reviewer for these helpful comments. The conclusion section now benefits from the inclusion of highlighting that the findings led to the development of the initial practice guidance for delivering Adolescent-Focused Low-Intensity LSW.

Introduction:

1. I'm not sure how well "looked after by social care" or "living in care" translates to other countries, so it would be worth expanding this in the opening paragraph, for example that these children are/have been in the care of their local authority, often in foster care etc.

- Author's response – We thank the reviewer for highlighting this. We have addressed this point using the phrase out-of-home care before then referring to adolescents in care.

2. You give a good description of what low intensity LSW entails, but providing details of what HI-LSW looks like by comparison would make the distinction clearer, both in terms of the therapeutic process,

and how it differs in the time spent on it... basically, what are the key features that differentiate HI and LI, and how do these differences impact outcomes? I know that this difference isn't a key feature of your work, but a few lines on this would really highlight the importance of good research on LI-LSW.

- Author's response – Thank you for your comments and suggestions. With this in mind and in line with Reviewer 1's comments, we have changed the structure of the introduction to allow easier comparisons between high and low intensity approaches to LSW.

3. Along similar lines, you mention that older children/adolescents often miss out, both in guidance and practice. But what makes LSW so important for adolescents? I imagine this will be to do with the level of insight and understanding that they have compared to younger children, and that the onset of most mental health conditions is in the teenage years (you do mention that the lack of guidance might influence the mental health inequalities, so this might be an appropriate juncture to add this), hence why a whole-childhood approach is necessitated... again, this will develop your rationale, making it abundantly clear to the reader why you are focussing on adolescents, that goes beyond "there isn't much for them".

- Author's response – Thank you to the reviewer for highlighting such important factors to strengthen the rationale for the need for this research to focus on adolescence. We have revised this and added to the rationale.

Methods:

1. Another quick point relating to rationale. Your first paragraph in this section is a little underdeveloped. Some unpacking of the contextual complexity of application of AF-LI-LSW, beyond adding a couple of citations, would help. What contexts has the literature explored? Under what circumstances might it work or not? What elements of this have not been explored, hence why you are conducting your study?

- Author's response – We thank the reviewer for their comments and have expanded this paragraph to clarify the rationale further.

2. A well-selected and well-explained methodology. I had never heard of this type of review before, but it's definitely been added to my mental list, thank you for that!

- Author's response: We thank the reviewer for their kind comments.

3. Please include, somewhere, an example search strategy (including terms), and how you devised these. Apologies if I missed this somewhere.

- Author's response – We have included a new section 'Data sources searched' which refers to Supplementary File 1 where we include our precise full search strategy as requested by Editors comments to Authors

4. You haven't provided any ethical procedures concerning the expert group. This is especially important given that some of them were vulnerable young people. Please describe the ethical approval you got, or else, how you decided this wasn't needed. Please also briefly describe the procedures of recruitment and consent that you undertook.

- Author's response – We thank the reviewer for highlighting this. We liaised with colleagues on the ethics board at the University of East Anglia prior to the commencement of the project as to whether ethical approval would be required for our expert consultants. As the Content Expert Groups were providing their consultation taking a PPI role as opposed to being participants of the research it

was advised that formal ethical approval would not be necessary. For such reasons, we have not provided procedures of recruitment or consent.

Results:

1. What criteria did you use to conclude that the evidence base was low quality?

- Author's response – In the paper's 'Limitations' section we highlight that we use the hierarchy of evidence for therapeutic studies (Evans, D., Hierarchy of evidence: a framework for ranking evidence evaluating healthcare interventions. *Journal of Clinical Nursing*, 2003. 12(1): p. 77-84).

2. Overall, though, a detailed results section, where links with literature and your groups were clearly made. Nicely done.

- Author's response: We thank the reviewer for their kind comments.

Discussion:

1. "Existing literature was mainly qualitative in nature". This isn't a valid limitation without further explanation. Quite the contrary, at first glance I can only see how qualitative work is an asset within the field you're looking at (your study focusses on nuance and complexity, which can't always be gauged from statistics). Whilst this may be my ignorant view, you nevertheless need to say why this is a limitation, i.e., how would the presence of more quantitative studies have mitigated this limitation? What else would such studies have provided your review?

- Author's response: We thank the reviewer for their comments. We have expanded this section to include comments which address the point from a realist perspective. The key point being that, working in one data medium method rather than being multi-method is a common shortcoming of realist works and one which we need to acknowledge. We have unpacked this and realise this should have been more overt.

2. "Silver bullet" – avoid using metaphors or sayings, as they aren't necessarily understood by people for whom English isn't their first language. Perhaps change to "magical solution" or "simple fix"?

- Author's response: We thank the reviewer for their recommendations and have added "magical solution to this complicated area of need"

3. I think your summary in supplementary file 4 is worth referring to at another juncture, perhaps at the beginning of the results section. This is a really nice condensed summary that someone might prefer to read if they are just skimming through your paper, so signposting it there would be useful.

- Author's response: We thank the reviewer for their recommendations and have signed posted as requested. This is now labelled as Supplementary File 3 as it comes before the previous Supplementary file in the paper now

Reviewer: 1

Competing interests of Reviewer: none

Reviewer: 2

Competing interests of Reviewer: None

VERSION 2 – REVIEW

REVIEWER	Banwell, Emily The University of Manchester, Manchester Institute of Education
REVIEW RETURNED	24-Jul-2023
GENERAL COMMENTS	Thank you for addressing the comments.